nanotechnology/quantum physics/solid state physics

field electron emission, empirical field emission equation, pre-exponential voltage exponent, field electron emission theory–experiment comparisons, least-residual method, local-gradient method

**Authors for correspondence:**
R. G. Forbes
e-mail: r.forbes@trinity.cantab.net
E. O. Popov
e-mail: e.popov@mail.ioffe.ru

# The pre-exponential voltage-exponent as a sensitive test parameter for field emission theories

R. G. Forbes[1], E. O. Popov[2], A. G. Kolosko[2] and S. V. Filippov[2]

[1]Advanced Technology Institute and Department of Electrical and Electronic Engineering, University of Surrey, Guildford, Surrey GU2 7XH, UK
[2]Ioffe Institute, ul. Politekhnicheskaya 26, Saint Petersburg 194021, Russia

 RGF, 0000-0002-8621-3298

For field electron emission (FE), an empirical equation for measured current $I_\mathrm{m}$ as a function of measured voltage $V_\mathrm{m}$ has the form $I_\mathrm{m} = CV_\mathrm{m}^{\,k}\exp[-B/V_\mathrm{m}]$, where $B$ is a constant and $C$ and $k$ are constants or vary weakly with $V_\mathrm{m}$. Values for $k$ can be extracted (i) from simulations based on some specific FE theory, and in principle (ii) from current–voltage measurements of sufficiently high quality. This paper shows that a comparison of theoretically derived and experimentally derived $k$-values could provide a sensitive and useful tool for comparing FE theory and experiment, and for choosing between alternative theories. Existing methods of extracting $k$-values from experimental or simulated current–voltage data are discussed, including a modernized 'least residual' method, and existing knowledge concerning $k$-values is summarized. Exploratory simulations are reported. Where an analytical result for $k$ is independently known, this value is reliably extracted. More generally, extracted $k$-values are sensitive to details of the emission theory used, but also depend on assumed emitter shape; these two influences will need to be disentangled by future research, and a range of emitter shapes will need examination. Other procedural conclusions are reported. Some scientific issues that this new tool may eventually be able to help investigate are indicated.

## 1. Introduction

Field electron emission (FE) is one of the paradigm examples of quantum-mechanical tunnelling, and has importance as such. FE is also important in technological contexts, as the emission mechanism of the electron sources used in many machines of

nanotechnology, including electron microscopes, and as one of the mechanisms contributing to undesired vacuum breakdown in many contexts, in particular the development of new high-gradient particle accelerators at organizations such as CERN. There is a strong case for having good detailed understanding of FE theory.

For some years, the Ioffe Institute FE research group has studied the experimental behaviour of 'CNT LAFEs', i.e. large-area field electron emitters (LAFEs) formed using carbon nanotubes (CNTs) partially embedded in a supporting matrix (e.g. [1–3]). The original motive for the work reported here was to improve the understanding of how FE theory applies to CNT LAFEs, and how it could be used to extract information from experimental measurements. In particular, there is technological interest in defining and extracting the effective emission area, which is only a small fraction of the LAFE macroscopic area (or 'footprint'). However, we have come to realize that more basic scientific problems need solution first. This paper begins to explore these problems. As is common in FE literature, this paper uses the so-called electron emission convention, whereby field, current density and current are taken as positive quantities that are the negatives of the corresponding quantities as derived in conventional electrical theory. Where stated, values of universal constants used in FE theory are given to seven significant figures.

A convenient starting point is Murphy–Good (MG) FE theory [4,5], which was developed in 1956 in order to correct an error in the original 1928 FE theory of Fowler and Nordheim. MG FE theory is an established 'mainstream' approach that (i) disregards the existence of atomic structure, and (ii) models field emitters as Sommerfeld-type free-electron conductors with a smooth planar surface of large lateral extent. Tunnelling takes place, from electron states near the emitter Fermi level, through a barrier that is assumed to be a planar image-rounded barrier—often now called a *Schottky–Nordheim (SN) barrier* [5].

The finite-temperature core form of the MG FE equation gives the local emission current density (LECD) $J_L^{MGT}$ in terms of the local work function $\phi$, the emitter temperature $T$ and the magnitude $F_L$ of the local surface electrostatic field ($F_L$ is called here the *local barrier field*). The equation for $J_L^{MGT}$ can be written in the linked form

$$J_L^{MGT} = \lambda_T t_F^{-2} J_{kL}^{SN} \tag{1.1a}$$

with

$$J_{kL}^{SN} \equiv a\phi^{-1} F_L^2 \exp\left[\frac{-v_F b\phi^{3/2}}{F_L}\right]. \tag{1.1b}$$

Here, $a$ and $b$ are the Fowler–Nordheim (FN) constants ([6]; see the electronic supplementary material for information about universal constants used in FE); $v_F$ and $t_F$ are appropriate particular values of the field emission special mathematical functions $v(x)$ and $t_1(x)$, as discussed in appendix A; and $\lambda_T$ is a temperature-dependent correction factor as defined in appendix A. $J_{kL}^{SN}$ is defined by equation (1.1b) and is termed the (local) *kernel current density for the SN barrier*. As demonstrated below, all of $v_F$, $t_F$ and $\lambda_T$ are dependent on the barrier field.

Some years ago [7], a simple good approximation was found for $v_F$. In terms of $F_L$, this takes the form [7,8]

$$v_F \approx 1 - \left(\frac{F_L}{F_R}\right) + \left(\frac{1}{6}\right)\left(\frac{F_L}{F_R}\right)\ln\left(\frac{F_L}{F_R}\right). \tag{1.2}$$

Here, the *reference field* $F_R$ is the barrier field necessary to pull the top of an SN barrier of zero-field height $\phi$ down to the Fermi level, and is given (originally, in older form, in table 1 in [9]) by

$$F_R = c^{-2}\phi^2 \cong (0.6944615 \text{ V nm}^{-1})\left(\frac{\phi}{eV}\right)^2, \tag{1.3}$$

where the *Schottky constant* $c$ [6] is given by $c = (e^3/4\pi\varepsilon_0)^{1/2}$, where $e$ is the elementary positive charge and $\varepsilon_0$ is the vacuum electric permittivity. For illustration, $\phi = 4.600$ eV yields $F_R \approx 14.695$ V nm$^{-1}$.

Substitution of equation (1.2) into equation (1.1b), and algebraic manipulation, leads to the good approximation

$$J_{kL}^{SN} \approx a\phi^{-1}(\exp \eta) F_L^{(2-\eta/6)} \exp\left[\frac{-b\phi^{3/2}}{F_L}\right], \tag{1.4}$$

where $\eta$ is a dimensionless work-function-dependent parameter defined by

$$\eta(\phi) \equiv bc^2\phi^{-1/2} \cong 9.836239\left(\frac{\text{eV}}{\phi}\right)^{1/2}. \tag{1.5}$$

For illustration, $\phi = 4.600$ eV yields $\eta \approx 4.5862$. The expression $(-\eta/6)$ will re-emerge as a contribution to the parameter $k$ in equation (2.1).

For a practical FE device/system, the relationship between $F_L$ and the measured voltage $V_m$ can be written

$$F_L = \frac{V_m}{\zeta_L}, \tag{1.6}$$

where $\zeta_L$ is the local *voltage conversion length* (VCL). An FE device/system is termed *ideal* if (i) the measured current $I_m$ is equal to the emission current $I_e$, (ii) emitter surface work functions can be treated as effectively constant, and (iii) at every surface location 'L', the VCL $\zeta_L$ can be treated as constant, independent of the values of $F_L$, $V_m$ and $I_m$. Obviously, the value of $\zeta_L$ depends on location.

Many practical FE devices/systems are non-ideal, over all or part of their range of operation, owing to 'complications', such as (among others) series resistance in the current path from the emitter to the high-voltage generator, current dependence in $\zeta_L$, heating-induced changes in work function, space-charge effects, and/or field dependence or time-dependence in emitter geometry. This paper deals only with ideal FE devices/systems.

Real emitters are often post-shaped or needle-shaped, with rounded ends, or are otherwise 'pointy'. We call such emitters *point-form emitters*. For an ideal FE device/system involving a point-form emitter, if some expression $J_L$ for LECD is available, and the local barrier field $F_L$ (and any other relevant parameters) are known at all relevant surface positions on the emitter, then the total emission current $I_e$ (and hence the measured current $I_m$) are found by integrating the LECD over the emitter surface 'S', and writing the result in the form

$$I_m = I_e = \int_S J_L dS \equiv A_n(F_a) \cdot J_a(F_a). \tag{1.7}$$

The LECD apex value $J_a(F_a)$ is a function of the apex barrier field $F_a$, and possibly of other relevant parameters not shown explicitly, such as the apex radius of curvature. The parameter $A_n$ is defined by this equation and is termed the *notional emission area*. Old work [10] suggests that $A_n$ is expected to be a function of $F_a$, rather than constant, and this is confirmed by recent theoretical analyses (see [11, pp. 406–408], and also [12]), as well as by the simulations discussed below. $A_n$ may also depend on other relevant parameters.

For ideal FE devices/systems, these theoretical dependences on $F_a$ convert directly into corresponding experimental dependences on the measured voltage $V_m$, via the formula $F_a = V_m/\zeta_a$ (where $\zeta_a$ is the constant apex VCL), and we can write

$$I_m(V_m) = A_n(V_m) \cdot J_a(V_m). \tag{1.8}$$

Equation (1.1) above applies to FE from smooth, flat, planar structure-less surfaces. Strictly, when dealing with tunnelling from curved surfaces, it is necessary to replace $v_F$ in equation (1.1*b*) by some more general barrier-form correction factor. However, for simplicity in our initial explorations, the present work uses the *planar transmission approximation*. For equation (1.1), this means that, in the integration in equation (1.3), we use—at any surface position 'L'—the value of $J_L^{\text{MGT}}(F_L)$ that corresponds to the local barrier field $F_L$ at 'L' (and similarly for other LECD formulae strictly valid only for planar surfaces).

The planar transmission approximation is not, in fact, a good approximation if the emitter apex radius is 'small' (often taken to mean 'less than 10 to 20 nm'—though recent work [12] suggests that 'less than 100 nm', or even 'less than 1 μm', might be a better criterion). However, in this paper, the planar transmission approximation is acceptable, because it allows us to start exploring how emitter shape affects the integration process.

Partly because there is non-integral field dependence in the pre-exponential in equation (1.4), and partly because there is expected to be voltage dependence in $A_n$ as well as in $J_a$ (but mainstream FE theories usually take $A_n$ as constant), the present work needs to employ a current–voltage FE equation different in form from those normally used. Hence, we use the so-called *empirical FE equation*

$$I_m = CV_m{}^k \exp\left[\frac{-B}{V_m}\right], \tag{1.9}$$

where $k$ is the *pre-exponential voltage exponent*, and $C$ and $B$ are in the first instance treated as constants, at least over limited ranges of measured voltage and current. (The form of equation (1.4) shows that this could be a legitimate first approximation, at least for present purposes.)

Equations of form (1.9) were first used in the early days of field electron emission, in particular by Abbott & Henderson [10], but only for integral values of $k$. Equation (1.9) was first used with a non-integral value of $k$ by Forbes in [8] (but with the symbol $\kappa$ instead of $k$).

The issues of interest in the present work are the theoretically expected behaviour and values of $k$, and methodologies for extracting reliable values of $k$ from experiments and simulations relating to current–voltage characteristics (IVCs). Predicting the behaviour and value of $k$ seems to involve both relatively straightforward questions and very deep physical questions, well outside the scope of the present paper. The aim here is to begin to address some of the simpler issues, primarily by means of simulations based on variants of the MG FE equation. What we shall show in the present paper is that the value of $k$ is affected by both the details of LECD theory and by the details of the integration over emitter shape, and that the value of $k$ seems to depend quite sensitively on details of LECD theory.

This opens the possibility that, if effects owing to LECD theory and effects owing to integration over emitter shape can be disentangled, then measurement of the value of $k$ might in principle provide a useful new tool for testing or distinguishing between different LECD theories. At the present time, there seem to be few experimental results available that have sufficiently high quality to be useful. Our hope is that a demonstration, by means of simulations, of what seems a useful new tool will in due course stimulate the generation of high-quality experimental results, and thus facilitate FE theory–experiment comparisons that are more precise and more reliable than has historically been the case.

The purpose of the present paper is to establish a 'proof of concept'. Detailed investigation of how the methods discussed here might be applied to the advanced modern treatments of FE theory that can now be found in the literature needs to be the subject of future research.

The paper's structure is as follows. Section 2 develops background FE theory further, presents existing predictions for the value of $k$ and discusses methods for extracting $k$-values from simulated or experimental data. Section 3 tests two of these extraction methods by means of simulations on planar emitter models; §4 then tests the methods using simulations on the so-called hemisphere-on-cylindrical-post (HCP) model for a CNT standing on a planar plate of large lateral extent. Section 5 looks again at the wider issue of how to compare FE experiment and theory, and outlines procedural conclusions reached in the present work. Section 6 provides a brief overall summary, indicates some experimental issues that will require attention and indicates some of the scientific issues in FE theory that this new tool may eventually be able to help investigate.

A preliminary account of this work was given as a conference poster in July 2019 [13].

# 2. Background theory

## 2.1. Aspects of empirical current–voltage theory

When written in the so-called *power-k coordinates*, equation (1.9) becomes

$$\ln\left\{\frac{I_{\mathrm{m}}}{V_{\mathrm{m}}{}^{k}}\right\} = \ln\{C\} - \frac{B}{V_{\mathrm{m}}}. \tag{2.1}$$

A data plot of this form is called here a *power-k plot*. For ideal FE devices/systems, a power-k plot can generate straight (or nearly straight) lines, and hence a constant or nearly constant value for $B$, whatever value is chosen for $k$ in the range of present potential interest, namely $-1 \le k \le 4$.

However, questions exist about expected values for $k$. There has been intermittent discussion (still largely unresolved) for many years. For simplicity in this initial treatment, we shall limit detailed discussion to situations where there is only one emitter, to temperatures near room temperature and below, and make the usual assumption that (as a first approximation) the local work function can be taken as uniform across the emitter surface. However, we first set out a slightly more general theoretical discussion.

To discuss theoretical predictions (for an ideal FE device/system), it is convenient to start from four linked equations that define a slightly generalized FE equation for $I_{\mathrm{m}}$ in terms of $V_{\mathrm{m}}$, namely

$$I_{\mathrm{m}} = A_{\mathrm{n}} J_{\mathrm{a}}, \tag{2.2a}$$

$$J_{\mathrm{a}} = \lambda J_{\mathrm{ka}}, \tag{2.2b}$$

$$J_{\mathrm{ka}} = Z_{\mathrm{F}}^{\mathrm{el}} D_{\mathrm{Fa}} = (a\phi^{-1}F_{\mathrm{a}}{}^{2})\,D_{\mathrm{Fa}} = (a\phi^{-1}\zeta_{\mathrm{a}}^{-2}V_{\mathrm{m}}{}^{2})\,D_{\mathrm{Fa}}, \tag{2.2c}$$

$$D_{\mathrm{Fa}} = \frac{1}{1 + \exp[G_{\mathrm{Fa}}]} \approx \exp[-G_{\mathrm{Fa}}], \tag{2.2d}$$

and
$$G_{\mathrm{Fa}} = \frac{\nu_{\mathrm{F}}b\phi^{3/2}}{F_{\mathrm{a}}} = \frac{\nu_{\mathrm{F}}b\phi^{3/2}\zeta_{\mathrm{a}}}{V_{\mathrm{m}}}. \tag{2.2e}$$

In these equations, the symbol $D$ denotes tunnelling probability. In principle, this is being evaluated here using the Kemble formalism [14], but our simulations in this paper will always be done for parameter ranges where this reduces (as shown) to the usual simple-Jeffreys-Wentzel-Kramers-Brillouin (JWKB) formalism [14]. Equation (2.2$d$) is an expression for the apex value of this tunnelling probability, for an electron with normal-energy level equal to the Fermi level, and equation (2.2$e$) provides a definition of the related apex barrier strength $G_{Fa}$. The parameter $\nu_F$ (nu$_F$) is a barrier-form correction factor that relates to the form of tunnelling barrier seen by this electron, and $\zeta_a$ is the apex VCL.

In equation (2.2$c$), the apex kernel current density $J_{ka}$ is given as a product of $D_{Fa}$ and the bracketed term $(a\phi^{-1}\zeta_a^{-2}V_m^2)$, which represents the 'effective incident current density' $Z_F^{el}$ (at the Fermi level), onto the inside of the tunnelling barrier, as found in elementary theory based on assuming an exactly triangular tunnelling barrier and $T = 0$ K.

In equation (2.2$b$), the LECD $J_a$ is given as the product of $J_{ka}$ and a correction factor $\lambda$ that accounts *formally* for various theoretical effects, including corrections owing to the use of atomic-level wave functions, corrections owing to the use of improved tunnelling theory and corrections to the effective incident current density, including corrections owing to improved integration over internal electron states and to temperature effects. Equation (2.2$a$) is equation (1.8).

In equations (2.2$a$)–(2.2$e$), the following symbols also refer to apex values, namely $\phi$, $\nu_F$, $Z_F^{el}$ and $\lambda$, but (for notational simplicity) this is not explicitly shown.

The generalized FE equation is written out in this way in order to display possible sources of voltage dependence in $k$, for ideal field-emitting systems. Strong dependence comes from the $V_m^2$ term in equation (2.2$b$), and significant dependence can come from the factor $\nu_F$ in equation (2.2$e$) and from the area $A_n$ in equation (2.2$a$). Weak dependence (usually neglected) can come from field dependence in the local work function $\phi$, though more so in the exponent in equation (2.2$e$) than in the pre-exponential in equation (2.2$c$). There will also be voltage dependence in the factor $\lambda$ in equation (2.2$b$), but the full likely strength of this dependence is not known. For non-ideal systems, there would or could be other significant sources of voltage dependence, not considered here.

This brief discussion supports two procedural conclusions: (i) the planar emitter case needs to be considered first, separately from the 'point-form-geometry case'; and (ii) emission from real post-like or needle-like emitters needs to be investigated by simulations that involve appropriate geometrical models. We next review existing knowledge concerning the value of $k$.

## 2.2. Review of previous discussions about the value of $k$

The original experimental discovery of approximate linearity between $\log_{10}\{I_m\}$ and $1/V_m$, by Lauritsen [15,16], and apparently independently by Oppenheimer (though unpublished at the time), is equivalent to $k = 0$. The original (1928) FN theoretical treatment [17] (i) disregarded atomic structure, (ii) used a smooth planar emitter model, (iii) is based on the assumption of an exactly triangular (ET) barrier, (iv) took the emitter temperature $T$ as zero, and (v) is obtained from equation (2.2) by setting: $\nu_F = 1$ and $\lambda$ equal to the FN pre-factor $P_F^{FN}$ [6] (which is independent of field and voltage). As is well known, the predicted $k$-value is 2. In 1929, Stern et al. [18] concluded that (in the one case where sufficiently sensitive experimental data existed) $k = 2$ gave better linearity than $k = 0$.

If the planar transmission approximation is used to apply the original 1928 FN FE theory to a point-form emitter, then the predicted 'total' value $k_t$ of $k$ is $2 + k_A$, where $k_A$ is the contribution made by any dependence of $A_n$ on apex field $F_a$ and hence voltage $V_m$. In 1939, Abbott & Henderson (AH) [10] re-analysed existing experimental results and found the best empirical value was $k_t = 4$, though they also suggested that if the difference from the FN 1928 result were associated with voltage dependence in emission area, then $k_t = 3$ was the simplest theoretically expected result, since a Taylor-expansion approach led to the result $k_A = 1$.

The SN barrier that includes the classical image potential energy (PE) is better physics [19] than the ET barrier. The SN barrier was first used in FE theory in 1928 by Nordheim [20], but a mathematical mistake was made (see [19]). As already indicated, MG introduced corrected equations in 1956 [4,5]. The finite-temperature MG FE equation, given earlier as equation (2.1), is obtained from equation (2.2) by setting $\nu_F = v_F$ in equation (2.2$d$), and $\lambda = \lambda_T t_F^{-2}$ in equation (2.2$b$); the derivation of this finite-temperature FE equation is considered to be approximately valid within ranges of field and temperature shown in [4,21].

When compared with the (zero-temperature) original 1928 FN equation, there is additional field and voltage dependence in the functions $v_F$, $t_F^{-2}$ and $\lambda_T$. The voltage dependence in $v_F$ is particularly significant, as is obvious from the '$-\eta/6$' term in the field dependence in the pre-exponential in equation (1.4).

When $\phi = 4.600$ eV, then $-\eta/6 \approx -0.7644$. For a smooth planar emitter, this yields the predicted value $k_t = k_P \approx 1.236$; for a point-form emitter (using the planar transmission approximation) $k_t \approx 1.236 + k_A$. If the AH theory that $k_A = 1$ were correct, and if it were correct to disregard other sources of voltage dependence, then we might expect $k_t \approx 2.236$. [But, in fact, it is unlikely that either of these things are correct.]

There are also more recent discussions. These imply that the predicted value of $k$ may depend on the emitter shape (e.g. [11,12,22]), and on the range of voltages considered [1], and (for LAFEs) might be affected by electrostatic depolarization effects (commonly called 'shielding') [22].

## 2.3. Existing methods for extracting empirical values for $k$

In the literature, three methods have been suggested for determining $k$ experimentally. The first is the AH (or *least-residual* (LR)) *method* [10], as follows: (i) plots of the form of equation (2.1) are made, for various values of $k$; (ii) for each, a statistical parameter (the 'residual') is evaluated that assesses the degree of linearity of the related plot; and (iii) the plot found to be 'most linear' (by virtue of having the lowest residual) determines the value of $k$. AH used only integral values of $k$, but the theory above shows that non-integral $k$-values need to be considered [8]. The AH method derives from the earlier work by Stern *et al.* [18], noted above.

The second method was suggested by Forbes [8]. On assuming that $B$ and $C$ can be treated as constants, the use of equation (1.9) to differentiate $\ln\{I_m\}$ with respect to $1/V_m$ leads to

$$g_{F1} \equiv -\frac{d\ln\{I_m\}}{d\{1/V_m\}} = kV_m + B. \tag{2.3}$$

Alternatively, result (2.3) can be written in the equivalent form

$$g_{F2} \equiv \left(\frac{V_m^2}{I_m}\right)\frac{dI_m}{dV_m} = kV_m + B. \tag{2.4}$$

Hence, one expects that $k$ can be found from the slope of a plot of $g_{F1}$ or $g_{F2}$ against $V_m$. We refer to this as the *local-gradient* (LG) *method*.

In [8], practical implementation of the LG method met problems. The FE IVCs reported by Dyke & Trolan [23], for tungsten emitters in traditional field electron microscope geometry, are widely accepted as very careful measurements. However, the attempt [8] to re-plot them via equation (2.3) failed to yield a useful result, apparently owing to noise in the original data and/or in its processing. It was suggested in [8] that probably the best experimental approach would be via equation (2.4), with $dI_m/dV_m$ found directly by an experimental methodology that uses phase-sensitive detection techniques.

More recently [24], IVC measurements on an array of nickel nanorods have been plotted both as a *Millikan–Lauritsen* (ML) plot (i.e. a plot of the form $\ln\{I_m\}$ versus $1/V_m$ [15,25]), and as a plot of form (2.3). An apparent experimental value $k_{expt} = 1.82$ was extracted from the second plot. However, because the ML plot, although quite smooth, is markedly nonlinear, this result for $k_{expt}$ cannot be considered reliable.

A third method, related to that of [8], has been suggested by Zubair *et al.* [26]. In fact, they considered an alternative empirical formula (thought to better represent emission from rough surfaces), but their method can also be applied to equation (1.9) and yields

$$\lim\left(\frac{1}{V_m} \to 0\right)\frac{d\ln\{I_m/V_m^2\}}{d\ln\{V_m\}} = k - 2. \tag{2.5}$$

When their graphs are interpreted in this way, the results (for three datasets, taken from different emitter materials) are $k_{expt} = 0.8$, $k_{expt} = 2$, and 'data too noisy'. Unfortunately, because the method involves extrapolation to an axis well away from the data, the lever effect operates and significant uncertainty exists in the extracted numerical value. In principle, the Forbes LG method looks more useful.

# 3. Testing extraction methods: planar emitters

In the Ioffe FE Group experimental research, we attempted to apply both the LR method and the LG method to IVC data taken from CNT LAFEs, but encountered severe problems with experimental data noise. Sections 3 and 4 originated as attempts to validate/explore these extraction methods by

**Table 1.** Field electron emission equations used in the planar emitter simulations. (Each 'named' equation is defined by the approximations used for the factors $\lambda$ and $\nu_F$ in equation (2.2). The labels HP (high precision), F06 (Forbes 2006) and ES refer to the approximate formulae for $v(x)$ and $t_1(x)$ given in appendix A, and hence for the particular values $\nu_F$ and $t_F$. The labels 'MG0' and 'MG300' refer to the zero-temperature and finite-temperature ($T = 300$ K) versions of the MG FE equation, and the label 'KernelSN' to equation (3.2).)

| test | equation name | $\lambda$ | $\nu_F$ |
|------|---------------|-----------|---------|
| 1 | elementary | 1 | 1 |
| 2 | MG0 (ES, ES) | 1/1.1 | 0.95–1.03$f$ |
| 3 | KernelSN (F06) | 1 | $\nu_F$(F06) |
| 4 | MG0 (F06, F06) | $t_F^{-2}$(F06) | $\nu_F$(F06) |
| 5 | KernelSN (HP) | 1 | $\nu_F$(HP) |
| 6 | MG0 (HP, HP) | $t_F^{-2}$(HP) | $\nu_F$(HP) |
| 7 | MG300 (F06, F06) | $\lambda_T \times t_F^{-2}$(F06) | $\nu_F$(F06) |

means of simulations: current densities or currents are predicted using defined equations, and these predictions are then used as the inputs to the $k$-value extraction procedures.

Both methods have been tested for the following two geometries: (i) the simple (but usually unrealistic) planar emitter case; and (ii) the case in which a CNT is represented by a common literature emitter model, namely the 'HCP' model, where the post is standing upright on one of a pair of well-separated parallel planar plates. Sections 3 and 4 discuss the simulation results for these two cases. In all simulations, the local work function was taken as constant across the emitter surface, and equal to 4.60 eV.

In this part of the paper, in MG FE theory, it is convenient to use as the independent variable the *scaled field f* defined by

$$f \equiv \frac{F_L}{F_R} = c^2 \phi^{-2} F_L. \tag{3.1}$$

In the literature, this parameter $f$ has also been called the 'scaled barrier field' and also the 'dimensionless field'. In terms of $f$, equation (1.1*b*) can be re-written in so-called scaled form as [27]

$$J_{kL}^{SN} \equiv \theta f^2 \exp\left[\frac{-\nu_F \eta}{f}\right], \tag{3.2}$$

where $\eta$ is given by equation (1.5), as before, and $\theta$ is a work-function-dependent parameter given by

$$\theta(\phi) \equiv ac^{-4}\phi^3. \tag{3.3}$$

For illustration, $\phi = 4.600$ eV yields $\theta = 7.236 \times 10^{13}$ A m$^{-2}$. Appropriate values for the parameters $\nu_F$ and $t_F$ are obtained from the formulae in appendix A, by setting the mathematical variable $x$ (the Gauss variable) equal to the scaled field $f$, when using any particular chosen mathematical approximation for $v(x)$. A formal justification for this procedure is given in [28, pp. 429–431].

In the planar emitter case, the voltage conversion length $\zeta_a$ can be put equal to a suitable separation $d_{sep}$ between parallel plates (we took $\zeta_a = 200$ nm). The various theoretical approximations used for the LECD $J_L$, together with a 'short name' for the related equation, are shown in table 1.

Our implementation of the LR method was then as follows. First, a suitable voltage range was decided. In practice, we normally used a voltage range corresponding to values of the apex scaled field $f_a$ [$=F_a/F_R$] in the range $0.15 \leq f_a \leq 0.45$, which is the 'Pass' range for the orthodoxy test [27]. Then, for an ideal system, with a related set of predicted $J(V_m)$ data based on a chosen equation, a data plot of the form $\ln\{J/V_m^k\}$ versus $1/V_m$ was built. Typically, about 1000 data points were used. A straight line was fitted to this data plot by 'least squares' methods. The residual $R(k)$ equal to the 'sum of least square deviations' is used as a measure of linearity. For a given dataset, this procedure is carried out for numerous different values of $k$, and the value of $k$ that gives the lowest value of $R(k)$ is identified. The online data processing programme allows the range (e.g. $-5 \leq k \leq +5$) and the step-length of $k$-values to be set. The outcome is a parabolic-like plot of $R(k)$, as shown in figure 1*a*, from which the $k$-value corresponding to the minimum value of $R(k)$ is easily established.

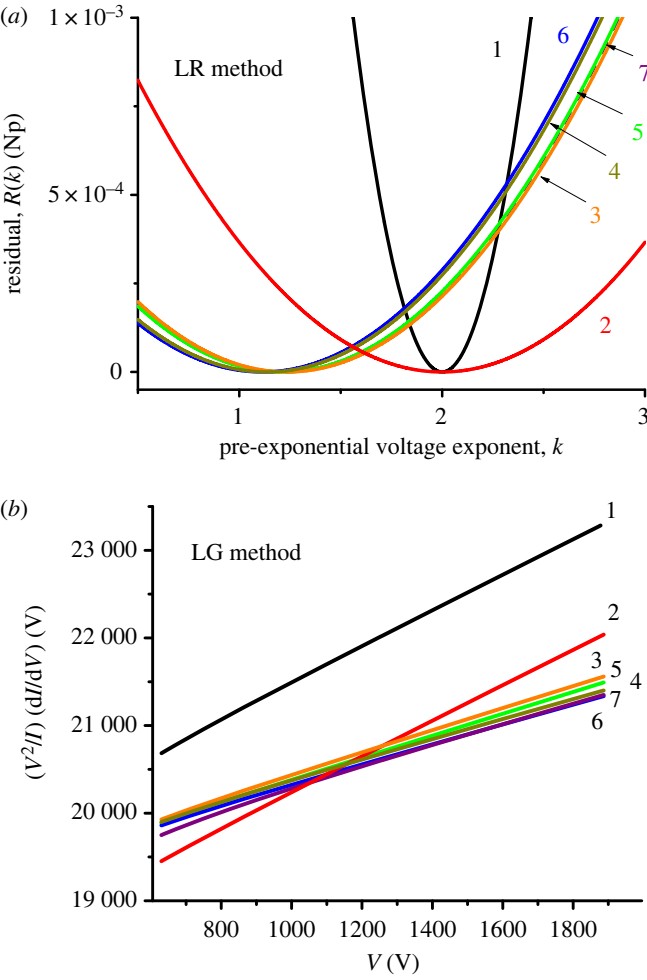

**Figure 1.** Simulated determinations of the value ($k_p$) of $k$ for a planar emitter by ($a$) the LR method and ($b$) the LG method, using the various equations named in table 1 to generate input data. The numerical labels correspond to the test numbers in table 1. With these simulations, $V$ denotes the input voltage and $I$ the related derived current. The deduced values of $k_p$ are shown in table 2.

The software package LABVEW was used to carry out these calculations.

As shown in table 1, seven different formulae were used to generate current–voltage datasets for our simulations and tests. Test 1 is based on the so-called elementary FE equation, in which the correction factors $v_F$ and $t_F$ in equation (1.1) are each replaced by unity. Tests 2–6 are based on different approximate versions of the zero-temperature MG FE equation (obtained by setting $\lambda_T = 1$ in equation (1.1$a$)). Test 7 is based on a version of the finite-temperature MG FE equation that applies at room temperature, taken as 300 K. The outcomes from these tests are summarized in table 2.

Results relating to the LR method were as follows. With the first two tests, the expression used for the correction factor in the exponent generates no voltage dependence in the pre-exponential in equation (1.9), so the value $k_p = 2$ is expected and found. The first three tests indicate that the methodology looks capable of extracting values of $k$ that are mathematically accurate to at least three decimal places. Comparison of tests 3 and 5, or alternatively 4 and 6, suggests the difference in the $k$-value between the F06 approximation and the high-precision formula for $v_F$ is around 0.021 (i.e. smaller than 2%), with the high-precision formula yielding a marginally lower result. Comparison of tests 3 and 4, or alternatively tests 5 and 6, shows that the inclusion of the term $t_F^{-2}$ in the expression for LECD reduces $k_p$ by about 0.1, that is about 10%. Comparison of tests 4 and 7 shows that the inclusion of the room-temperature correction term increases $k_p$ by about 0.08, i.e. about 7%; this increase is higher than was expected.

Results from the LG method exhibit the same qualitative trends as do those for the LR method, but numerical results are systematically higher than the corresponding results from the LR method, by around 4%. The exact reason for this is not fully understood, but is assumed to be related to slight curvature in the plots shown in figure 1$b$.

**Table 2.** To show predicted values (where available) of the total 'planar' pre-exponential voltage-exponent $k_p$, and values extracted from simulations based on the LR and LG methods, for a planar emitter with uniform local work function 4.6 eV. (The equations used for LECD are specified in table 1. In order to show the apparent consistency of the extraction methodology, predicted and extracted values for the LR method are given to 5 decimal places.)

| test | LECD equation | predicted $k_p$ | derived $k_p$ (LR) | derived $k_p$ (LG) |
|---|---|---|---|---|
| 1 | elementary | 2.00000 | 2.00000 | 2.00 |
| 2 | MG0 (ES, ES) | 2.00000 | 2.00000 | 2.00 |
| 3 | KernelSN (F06) | 1.23564 | 1.23560 | 1.28 |
| 4 | MG0 (F06, F06) | n/a | 1.13519 | 1.17 |
| 5 | KernelSN (HP) | n/a | 1.21200 | 1.26 |
| 6 | MG0 (HP, HP) | n/a | 1.11320 | 1.15 |
| 7 | MG300 (F06, F06) | n/a | 1.21640 | 1.24 |

Specific conclusions from this set of results are as follows: (i) for 'good' (noise-free) experimental data—the LR method would probably be more precise than the LG method; (ii) certainly in this planar emitter case, the LR method seems capable of extracting $k$-values with a precision of at least three decimal places; (iii) as regards the LECD formulae, there seems little physical merit in using anything other than the high-precision formulae for evaluating $v_F$ and $t_F$ in simulations of this kind, although it is conceivable that computing resource limitations could make use of the F06 formulae preferable in particular circumstances; and (iv) the correction terms ($t_F^{-2}$ and $\lambda_T$) in the pre-exponential have notable effects on the value of $k$.

# 4. Testing extraction methods: hemisphere-on-cylindrical-post models

With real point-form emitters, the local work function $\phi$, barrier field $F_L$ and LECD $J_L$ all vary across the surface. For simplicity in modelling, $\phi$ is usually taken as constant; this is also done here.

To model CNT shape, the well-known 'HCP' model (e.g. [29–32]) is used, with the post standing on one of a pair of well-separated parallel planar plates.

Three-dimensional modelling of the field distribution over the HCP model surface was carried out using the COMSOL finite-element software package. The strategy was to calculate the electrostatic fields that apply to small surface segments with height $\delta h$, for the hemispherical cap and the top part of the cylindrical surface. The total number of segments used was 1000.

The HCP parameters used were similar to those for the single-walled (SWCNT) and multi-walled (MWCNT) CNTs used in our experiments. For SWCNTs, this is radius $r_c = 1$ nm and height $h_{post} = 2.5$ µm ($h_{post}/r_c = 2500$); for MWCNTs, this is radius 4 nm and length 5 µm ($h_{post}/r_c = 1250$).

The simulation box was taken as a flat-ended cylinder with radius $r_{sim} = 250$ µm in both cases, and with height $h_{sim}$ given by

$$h_{sim} = 300 \ \mu m + h_{post}. \tag{4.1}$$

Because the dimensions of the simulation box are very much greater than those of the emitter, depolarization effects due to the emitter images in the sides and top of the box will be negligible, as shown by de Assis & Dall'Agnol [32].

In parallel-planar-plate geometry, the (dimensionless) local field enhancement factor (FEF) $\gamma_{PL}$ is given adequately by

$$\gamma_{PL} = \frac{F_L d_{sep}}{V_P}, \tag{4.2}$$

where $d_{sep}$ is the separation of the plates (here identical with the simulation box height $h_{sim}$), and $V_P$ is the voltage between the counter-electrode (anode) and the vacuum-facing surface of the emitter substrate. For an ideal FE system, $V_P$ is equal to the measured voltage $V_m$.

Figure 2 shows values of $\gamma_{PL}$, as calculated using finite-element methodology, as a function of the distance $L$ from the emitter apex, measured 'along the surface' in a plane that includes the emitter

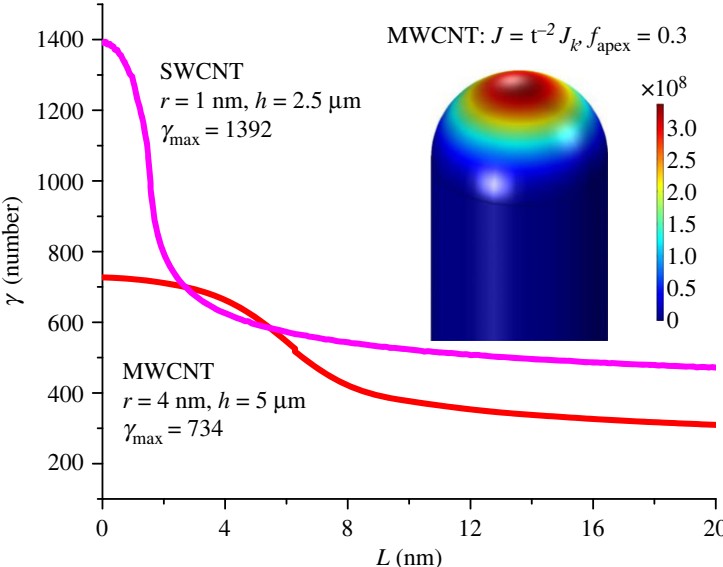

**Figure 2.** Calculated values of the local surface FEF $\gamma_{PL}$ as a function of the distance $L$ from the emitter apex, measured 'along the surface' in a plane that includes the emitter axis. Calculations are shown for an HCP model post of height $h = 2.5$ μm and radius $r_c = 1$ nm ($h/r_c = 2500$ and apex FEF $\gamma_a = 1392$, which models an SWCNT), and for a post of height 5 μm and radius 4 nm ($h/r_c = 1250$, $\gamma_a = 734$, which models an MWCNT). For the MWCNT model, the inset shows the variation, along the emitter surface, of the local current density, in A m$^{-2}$, calculated using the MG0 (F06, F06) formula.

axis, for HCP models of (i) our SWCNTs and (ii) our MWCNTs. For the MWCNT, the inset shows how the related local current density distribution (as calculated using the MG0 (F06, F06) formula) varies along the surface, in the case where the scaled-field value at the emitter apex is $f_a = 0.3$.

Using the planar transmission approximation, and a chosen LECD formula, emission current contributions from each segment were calculated and summed. Thus, IVCs for HCP models representing SWCNT and MWCNT emitters were obtained.

To explore the LR method, as applied to these HCP models, we used the same LECD expressions and methodology as before, but excluding the elementary FE equation. The results are summarized in table 3. Related 'parabolic graphs' would add little to the discussion, and are not provided.

We also explored the LG method, using the same set of LECD expressions. The $k$-values derived from all the LG method plots are shown in table 3. The illustrative plots for tests 20, 21, 28 and 29 are shown in figure 3.

The most striking feature of the results in table 3 is that, with the exception of the Elinson–Shrednik (ES) approximation, all the various formulae give very similar results for $k_A$, namely a value close to 0.5. In every case, the result for the SWCNT dimensions is slightly less than that for the MWCNT dimensions, suggesting that the shape of the HCP emitter does have at least a slight effect. The comparison of tests 18 and 19 with 12 and 13 shows that the results at room temperature are only marginally different from those for zero-temperature. The comparison of tests 12 and 13 with tests 24 and 25, or tests 16 and 17 with 28 and 29, shows that essentially the same results are being obtained from the LR and LG methods.

Specific physical conclusions from these simulations on HCP models are as follows. (i) The effect of pointed-emitter shape is to increase the value of the pre-exponential voltage-exponent $k$, by an amount $k_A$ relative to the corresponding planar emitter case. This is equivalent to the dependence of the notional emission area $A_n$ on the apex value $F_a$ of local barrier field, and hence (for ideal emitters) to dependence on the measured voltage. For an emitter with the apex radius of curvature $r_a$, this can be represented in terms of the behaviour of a *notional apex area efficiency* $g_n$ defined by

$$g_n \equiv \frac{A_n}{2\pi r_a^2}. \tag{4.3}$$

The parameter $g_n$ represents the fraction of the surface area (of a hemisphere of radius $r_a$) that would be emitting if the current density within the emitting area were constant and equal to its apex value. Obviously, for the HCP model, $r_a$ is equal to the cylinder radius $r_c$.

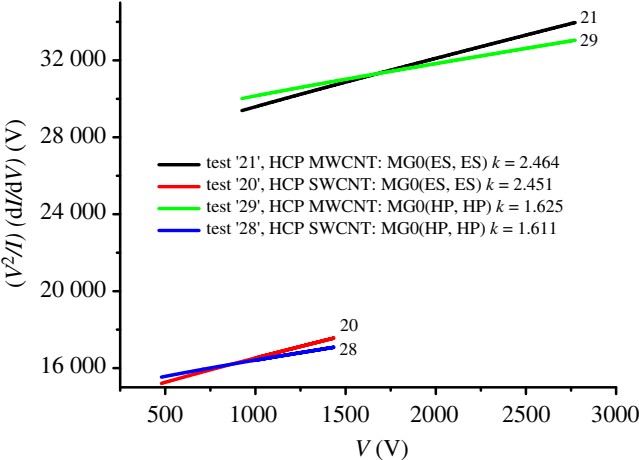

**Figure 3.** Illustrative results of investigations into using the LG method to determine estimates of the total voltage-exponent $k_t$, for HCP model emitters, using various different formulae for LECD, as specified in table 3.

**Table 3.** To show values derived for the total voltage-exponent $k_t$, for HCP models representing a single-walled CNT (SW) and a multi-walled CNT (MW). (Evaluation of the total emission currents assumes: (i) the local work function is uniform, with value 4.6 eV; (ii) the planar transmission approximation; and (iii) various different approximations, as defined in table 1, to the MG FE equation. Values of $k_t$ are extracted using both the LR method and the LG method. The columns labelled 'change' show the differences from the corresponding 'planar emitter' simulations.)

| CNT Type | LECD equation | $k_p$ from table 1 | least-residual method | | | local-gradient method | | |
|---|---|---|---|---|---|---|---|---|
| | | | test | derived $k_t$ | change, $k_A$ | test | derived $k_t$ | change, $k_A$ |
| SW | MG0 (ES, ES) | 2.000 | 8 | 2.455 | +0.455 | 20 | 2.451 | +0.451 |
| MW | MG0 (ES, ES) | 2.000 | 9 | 2.463 | +0.463 | 21 | 2.464 | +0.464 |
| SW | KernelSN (F06) | 1.236 | 10 | 1.731 | +0.496 | 22 | 1.728 | +0.492 |
| MW | KernelSN (F06) | 1.236 | 11 | 1.740 | +0.504 | 23 | 1.743 | +0.507 |
| SW | MG0 (F06, F06) | 1.135 | 12 | 1.637 | +0.502 | 24 | 1.631 | +0.496 |
| MW | MG0 (F06, F06) | 1.135 | 13 | 1.646 | +0.511 | 25 | 1.645 | +0.510 |
| SW | KernelSN (HP) | 1.212 | 14 | 1.707 | +0.495 | 26 | 1.707 | +0.495 |
| MW | KernelSN (HP) | 1.212 | 15 | 1.716 | +0.504 | 27 | 1.722 | +0.510 |
| SW | MG0 (HP, HP) | 1.113 | 16 | 1.614 | +0.501 | 28 | 1.611 | +0.498 |
| MW | MG0 (HP, HP) | 1.113 | 17 | 1.624 | +0.511 | 29 | 1.625 | +0.510 |
| SW | MG300 (F06, F06) | 1.216 | 18 | 1.719 | +0.503 | 30 | 1.699 | +0.481 |
| MW | MG300 (F06, F06) | 1.216 | 19 | 1.728 | +0.512 | 31 | 1.710 | +0.494 |

(ii) With the exception of the ES approximation, all approximate versions of the MG FE equation yield much the same value of $k_A$, which is in the vicinity of 0.5 (for the work function value 4.60 eV used). This is markedly less than the AH value of $k_A = 1$, and shows that their linear-Taylor-expansion approach is a poor approximation.

For the case of a hemisphere on a plane ($h_{post}/r_a = 1$), Jensen has developed an analytic expression for $g_n$ (see [11, pp. 407–408]); for a 4.50 eV emitter, his fig. 30.3 appears to indicate a value of $k_A$ near 0.8. For the case of a parabolic tip, theoretical analysis by Biswas [33] has found $k_A$ values close to unity. These results, together with our own finding of a slight difference between the SWCNT and MWCNT cases, tend to confirm that the value of $k_A$ depends in principle on the precise shape of the emitter—and that more extensive research investigations into shape effects are needed.

# 5. Discussion: the comparison of field electron emission theory and experiment

## 5.1. The historical situation

Making detailed comparisons between FE theory and experiment has long been a weak point of the subject area. Historically, the situation has been as follows. The good (albeit not precise) linearity of FN plots taken from ideal FE devices/systems establishes convincingly that FE is a wave-mechanical tunnelling process. However, the apparent linearity of an FN plot does not by itself provide useful information about the *details* of FE theory. In practice, it has proved extremely difficult to use comparisons between theory and experiment to distinguish decisively between different detailed FE models/theories.

For example, the FN (1928) and MG (1956) approaches to metal FE theory make significantly different physical assumptions about the nature of the tunnelling barrier. For the last 60 years or so, FE theoreticians have been in no doubt that the MG approach is 'better physics' [19]. However, an exploration of a significant part of the available experimental evidence some years ago [34] concluded that, although the balance of experimental evidence tended to favour the MG physical assumptions, the totality of the evidence examined did not allow this to be presented as a decisive conclusion. For example, temperature and field-dependent shifts in the peak of the total energy distribution (TED), which are certainly compatible [35] with the MG physical assumptions, also seem to be compatible [34] with the FN physical assumptions (within the accuracy of the published experimental TED data). This situation is a 'gap in the background of FE science' that deserves to be filled.

Unfortunately, the longstanding IVC analysis techniques based on the use of FN plots are not very helpful for making *basic* theory–experiment comparisons. For ideal FE devices/systems, the slope $S$ of a current–voltage-type FN plot basically gives information about the *electrostatics* of the FE device/ system in use, in that an extracted value of the apex VCL $\zeta_a$ can be obtained from the formula

$$\zeta_a^{\text{extr}} = \frac{-\sigma_t b \phi^{3/2}}{S}.$$

(5.1)

In MG FE theory, it is usually adequate to take the slope correction factor $\sigma_t$ as 0.95. In the original 1928 FN FE theory, $\sigma_t = 1$. Thus, any attempt to use the measured slope and the 5% difference in $\sigma_t$ to distinguish between these two theories needs a highly accurate independent determination of $\zeta_a^{\text{extr}}$; this would be very difficult in principle, and possibly impracticable in reality.

For ideal FE devices/systems, estimates of formal emission area $A_f$ can be obtained from the FN-plot intercept, most easily by using the relevant emission area extraction parameter (e.g. [36]). Estimates of $A_f$ derived, from a given experimental dataset, (i) by using MG FE theory and (ii) by using the original 1928 FN FE theory, can differ by a large factor, typically around 100. However, there are extraordinary difficulties in making a reliable theoretical prediction of the value of $A_f$, partly because of the limited information that currently exists about likely values of notional emission areas $A_n$, but more so because of the lack of good knowledge about the value and behaviour of the 'uncertainty factor' $\lambda$ in equation (2.2b).

It might be thought that the study of TEDs would provide a good comparison method. While there is certainly scope for further careful numerical investigations, initial explorations did not lead to convincing results, as reported above.

Against this background, studies of the pre-exponential voltage-exponent $k$ in the empirical FE equation, i.e. equation (1.9), appear to show significant promise as a tool for using experiments to distinguish between different detailed FE theories. The results in this paper should be understood as part of a preliminary exploration of the issues involved.

## 5.2. Summary of procedural conclusions

From our experience in carrying out simulations and comparisons, as described above, we conclude the following.

(1) At this point in time, theory and simulations need to work with ideal FE devices/systems, so that the complications associated with non-ideality can be avoided. Also, experiments should, as far as possible, be made on ideal devices/systems. The following remarks concentrate on ideal FE devices/systems.

(2) In the context of 'empirical' equation (1.9), when dealing with classical-conductor models of field emitters, it appears that the theory can assume (at least initially) that the parameter $B$ has a constant value, and that the issue is the behaviour of the pre-exponential $CV_m^k$.

(3) Comparison of equations (1.8) and (1.9) suggests that this pre-exponential $CV_m^k$ can, at least conceptually, be decomposed as a product of two components:

$$CV_m^k = C_1 V_m^{k_1} \cdot C_2 V_m^{k_2}. \tag{5.2}$$

The second term relates to the behaviour of a current density defined at the emitter apex, and the first to the behaviour of an appropriately defined notional emission area.

(4) Thus, we expect that the parameters $C$ and $k$ may both contain entangled information about the physics of the emission process and the shape of the emitter. In principle, what needs to be done is to untangle this information, in order to provide reliable information about the emission physics, the emitter shape and the effective emission area. This seems a research task that may prove complex and possibly lengthy. As things stand at present, it seems a higher priority to use extracted $k$-values to deduce conclusions about the emission physics, rather than to use extracted values of $k$ and $C$ to deduce conclusions about emission area.

(5) There seem some obvious initial avenues of exploration, as follows.

   (a) Within the context of the planar transmission approximation, we need to explore more comprehensively how emitter shape affects the value of $k$. We also need to understand how these results are affected by the value chosen for the local work function.

   (b) We need to establish a link between (i) the results of simulations and (ii) the analytical results of Jensen [11] and Biswas [33] concerning the notional apex area efficiency $g_n$. Further, we need to establish how $g_n$ depends on emitter shape and on the assumptions made about the emission physics (particularly the barrier form).

   (c) We need to investigate the behaviour of the shape contribution $k_A$ in the limit of large emitter apex radius. If $k_A$ turns out to be small (or, alternatively, very well defined), then we need to explore whether experimental measurements on 'blunt emitters' could provide decisive information about the emission physics for 'nearly planar emitters'. The mainstream approximation, as described at the start of this paper, involves the neglect of atomic structure, the use of the simple-JWKB tunnelling formalism and the assumption of an SN tunnelling barrier. As indicated above, it would be helpful to have it *decisively* confirmed experimentally that this is an 'adequate physical approximation' for a blunt emitter.

The following detailed procedural conclusions have also been reached.

(6) Given that the extracted value of $k_A$ depends relatively weakly on the precise approximate form used for the MG FE equations, it should be sufficient—in initial future explorations of the effects of emitter shape—to approximate the local current density by the kernel current density for the SN barrier.

(7) Given that our results for $k_t$ depend notably on the accuracy of the approximate formula used for $v_F$, there will be little physical merit, in future simulations, in using anything other than the high-precision formula for $v(x)$ given in appendix A (though, as noted earlier, computing resource limitations might indicate the use of the F06 approximation in particular circumstances).

(8) There seems limited merit in using the ES approximation in future simulations.

(9) The present simulations cover the scaled-field range $0.15 \leq f_a \leq 0.45$, on the grounds that ideal FE devices/systems will normally operate within this range. In this range, extracted values of $k$ appear (so far) to be nearly constant, and one would expect the LR and LG methods to return nearly the same numerical result. However, when making comparisons between simulations and analytical models (e.g. for $g_n$), it may be useful to carry out simulation and extraction procedures over a wider range of values within the range $0 \leq f_a \leq 1$. We have detected that, when considered over wider ranges of $f_a$-values, $k$ may need to be treated as a weakly varying function of $f_a$, and hence of measured voltage. In this case, the LR method yields a reasonably precise 'effective average value' of $k$, but the LG method can indicate whether any significant variations in $k$ exist as a function of voltage.

# 6. Summary and research outlook

## 6.1. The role of the present paper

This investigation started as an attempt to understand in more detail the IVCs derived from CNT-based LAFEs operating in industrial vacuum conditions. However, we came to realize that

more fundamental questions relating to the analysis of IVCs and the effect of emitter shape on IVCs need to be explored first. Further, experiments that seek to establish the value of the pre-exponential voltage-exponent $k$ might eventually open a path towards quantifying fundamental uncertainties known to currently exist in tunnelling theory, particularly as this applies to curved emitters.

By using the empirical FE equation, this paper has shown that comparisons of extracted $k$-values should provide a useful new tool for comparing FE theory and experiments. Simulations have shown that small theoretical variations, both in the barrier-form correction factor and in pre-exponential correction factors, can be detected, by the 'LR' and/or 'LG' methods. Using the planar transmission approximation, it has been shown that emitter shape makes a significant contribution to $k$, and that small shape differences may have a detectable effect: this is an unwelcome (though not unexpected) complication.

This paper has aimed to establish a 'basic proof of concept' for this methodology based on finding $k$-values. We believe that this has been achieved. The paper does not aim to provide full discussion either of theoretical details of how this methodology might be implemented, or of how appropriate apparatus and experimental procedures might be designed. However, we briefly comment below on possible ways forward.

To address the whole problem of developing and testing truly reliable FE theory, using $k$-value methodology, one must assess what contribution each part of existing theory (and each new development) makes to the total value of $k$. This seems likely to generate a complicated progressive process of making comparisons between theory and experiment, in order to reach definitive or provisional (with caveats) scientific conclusions, and is likely to involve careful design of experiments.

## 6.2. Research outlook: experimental issues

As is usual scientific practice, initial experiments need to have as few 'complications' as possible, at least until some basic facts are built up. Thus, one would work with emitters fabricated from good metallic conductors (so that the system is 'ideal'); one would work with single emitters rather than arrays, and would avoid using excessively sharp emitters; one would work at room temperature (or maybe below); and one would work within a voltage range that corresponds a scaled-field range well within the validity limits for MG finite-temperature FE theory.

It would be desirable to work with stable, clean emitting surfaces, and to avoid noise issues of all kinds. This indicates the need to operate in good ultrahigh vacuum, with appropriate surface cleaning techniques applied before experiments are begun. It also indicates the need to use computerized control of voltages, computerized measurement of currents and computerized recording of current–voltage data. Conceivably, as previously suggested [8], one might wish to explore using phase-sensitive detection techniques to measure how the quantity $g_{F2}$ in equation (2.4) varies with measured voltage.

Clearly, to avoid the complications associated with non-planar emitter shape, it would be desirable (if at all possible) to measure currents from a defined area on an atomically flat plane, across which the applied electrostatic field is as uniform as possible.

If an experiment of this kind proves impracticable, at least in the short term, one might use something approximating to a traditional field electron microscope configuration, with a probe hole that limits the current to that drawn from a well-defined crystallographic region. The shape of the emitter should be established by separate electron microscope experiments.

## 6.3. Research outlook: theoretical scientific issues

As discussed above, a longstanding FE problem is the desirability (for scientific completeness) of having a decisive *experimentally based* confirmation that 1956 MG FE theory (which takes exchange-and-correlation effects into account by using image PEs) is 'better physics' than the original 1928 FN theory (which did not). For an emitter with uniform local work function 4.60 eV, the predicted difference in $k$ is around 0.7, as noted earlier, which is much greater that the apparent intrinsic accuracy of the simulations (around 0.001, or better).

To solve this and other problems, one needs to identify the theoretical effects likely to make the greatest-magnitude contributions to $k$, and to determine/estimate related contribution sizes. Obviously, the biggest is the contribution '+2' that comes from summation over emitter electron states, using elementary theory. Two of the next largest contributions (namely, using the SN rather than the

ET barrier, and using non-planar emitter shapes) have already been discussed. There is a need for further simulation-based research into how k-values are affected by emitter shape and size, and by the value and behaviour of the local work function.

The introduction of emission models that involve the atomic wave functions of surface atoms (e.g. [37]) will probably also contribute significantly to k-values. As suggested by Oppenheimer long ago [38], field electron emission from a surface atom resembles the field ionization of that atom, and it is known [39, see Problem 1 on p. 295] that the accepted formula for the field ionization rate-constant of a hydrogen atom involves a pre-exponential field term of the form $F^{-1}$. However, a closer analogy may be the Kingham [40] calculations on the post-field-ionization of field-evaporated metal ions, which are of significant interest in atom probe microscopy. In both cases, the implied contribution to k would or could be –1, or something close to this. Revisiting the theory of these topics, while also keeping an eye on modern density-functional-theory techniques of modelling charged metal surfaces (e.g. [37]), seems a useful line of exploration.

Beyond this, and possibly of significantly more physical interest (though necessarily topics for later exploration), there will be situations where the differences between different FE theories might be expected to be somewhat smaller. In particular, one might expect a theory of FE from curved surfaces, when the relevant LECD formula is used with a point-form emitter, to lead to a k-value slightly or somewhat different from that found using the planar transmission approximation. However, there is an issue of whether some existing methods of predicting tunnelling probabilities for curved emitters are strictly correct physical approximations; this is because careful reading of the analysis of hydrogen-atom field ionization by Landau & Lifschitz [39] suggests that the simple-JWKB approximation is valid only if the relevant component of the electron wave function in effect obeys the one-dimensional Schrödinger equation as found when separation is made in Cartesian coordinates.

Further beyond this, there may be issues relating to the implementation and reliability of the more advanced FE theories that assess tunnelling probability by means of path-integral methods more sophisticated than those commonly used (for example, [41,42]), and/or assess effects (including 'coherent-emission effects') occurring with emitters with apex shapes that are very small and very sharp (e.g. [43]).

With these advanced modern theories, a suitable way forward would be for their originators to use them to generate predicted IVCs (or, where relevant, predicted plots of tunnelling probability versus field). These could then be analysed by the methods discussed above, to yield a predicted k-value or a predicted contribution to k.

More generally, some of the known problems in FE theory appear, at least in part, to be general problems associated with tunnelling theory in quantum mechanics. The late Prof. Marshall Stoneham (a former President of the UK Institute of Physics) considered that some of the most difficult unsolved problems in theoretical physics were in FE theory (A. M. Stoneham 2001, private communication to R.G.F.). There may be the hope that, eventually, future scientific developments in FE could be of wider scientific use.

Data accessibility. Computations described in this paper have been carried out using two standard commercial software packages. COMSOL™ has been used to carry out electrostatic modelling. LABVIEW™ has been used for: (i) calculation of emission current densities and emission currents; (ii) regression-type calculations that generate values of the pre-exponential voltage-exponent k; and (iii) implementation of the 'local gradient' data analysis method described in the main text. When COMSOL™ is used, a large 'internal' data file is generated by COMSOL™ and passed to LABVIEW™. Appropriate modelling information (as described in the main text) and appropriate sequences of input instructions have to be entered into the software packages, but there are no large data input files and there has been no need for low-level code to be written. It is assumed that researchers able to drive the software packages used (or packages with equivalent functionality), are able to generate the package input instructions needed to implement the procedures described in the main text or in the electronic supplementary material, and to generate associated diagrams. Additional details concerning the electrostatic modelling, beyond those provided in the main text, are provided as the electronic supplementary material.

Authors' contributions. E.O.P. and R.G.F. collaborated in directing the research and preparing the manuscript; A.G.K. and S.V.F. carried out the detailed calculations; S.V.F. and R.G.F. collaborated in preparing the electronic supplementary material; all authors contributed to research discussions.

Competing interests. We declare we have no competing interests.

Funding. We received no specific funding for this study.

Acknowledgements. The authors are grateful to Fernando F. Dall'Agnol (Federal University of Santa Catarina) for his help in COMSOL simulations.

# Appendix A. Detailed formulae

## A.1. The field electron emission special mathematical functions v(x), u(x) and t₁(x)

The field emission special mathematical functions are expressed here in terms of the Gauss variable $x$, i.e. the independent variable in the Gauss hypergeometric differential equation (HDE). By appropriate choice [28,44] of the constants in the Gauss HDE, this equation can be reduced to the defining equation found by Forbes & Deane [5], namely

$$x(1-x)\frac{d^2 W}{dx^2} = \frac{3}{16} W. \tag{A1}$$

Note that the symbol $l'$ used in [44] has been replaced here by the symbol $x$ now preferred; there is no change in the underlying mathematics.

The *principal field emission special mathematical function* v(x) is the particular solution of equation (A1) that satisfies the boundary conditions [40]

$$\mathrm{v}(0) = 1; \quad \lim_{x \to 0}\left\{\frac{d\mathrm{v}}{dx} - \frac{3}{16}\ln x\right\} = -\frac{9}{8}\ln 2. \tag{A2}$$

A formal analytical solution for v(x) is stated, and an exact series expansion derived, in [44].

Related special mathematical functions u(x) and t₁(x) are defined in terms of v(x) by

$$\mathrm{u}(x) \equiv -\frac{d\mathrm{v}}{dx} \tag{A3}$$

and

$$\mathrm{t}_1(x) \equiv \mathrm{v}(x) + \left(\frac{4}{3}\right)x\,\mathrm{u}(x). \tag{A4}$$

The typesetting convention used here is that (like 'sin') the symbols for special mathematical functions are typeset upright.

It can be shown [28] (though the proof is not trivial) that these mathematical functions are applied to MG FE theory by setting $x$ equal to the local scaled field $f$ defined by equation (3.1). In the main text, the simplified notation is used that $\mathrm{v}_F \equiv \mathrm{v}(x=f)$, $\mathrm{t}_F \equiv \mathrm{t}_1(x=f)$; in some equations $f$ has the emitter apex value $f_a$.

Historically, the pure mathematics of 'v' has been formulated in terms of the Nordheim parameter $y$ [$=+\sqrt{x}$]. However, the discovery of equation (A1) as the defining equation for 'v' has helped to show that this choice is inappropriate when discussing the *pure mathematics* of 'v': the natural mathematical variable is the Gauss variable $x$. The Nordheim parameter $y$, when defined as $+\sqrt{f}$, remains acceptable as an alternative *modelling* variable, but strong reasons can be given [28] for preferring to use the scaled field $f$ as the modelling variable when discussing FE IVCs and related topics, and it would make for a cleaner overall system if the use of the Nordheim parameter $y$ in such contexts were progressively discontinued.

## A.2. High-precision formulae (HP formulae)

High-precision (HP) approximation formulae are given in [5] for v(x) and u(x). Over the range $0 \le x \le 1$ (but not outside this range), the magnitude of the maximum error in these formulae is $8 \times 10^{-10}$. The HP formulae take the form below, where the constants have the values given in table 4:

$$\mathrm{v}(x) \cong (1-x)\left[1 + \sum_{i=1}^{4} p_i x^i\right] + x\ln x \sum_{i=1}^{4} q_i x^{i-1}, \tag{A5}$$

and

$$\mathrm{u}(x) \cong \mathrm{u}(1) - (1-x)\sum_{i=0}^{5} s_i x^i - \ln x \sum_{i=0}^{4} t_i x^i. \tag{A6}$$

## A.3. The simple good approximations (F06 formulae)

Simple good approximations for the FE special mathematical functions are based on the formula for v(x) discovered empirically and reported in [7] (where the modelling variable $f$ was used, rather than $x$). As noted earlier, this 'F06 formula' has the form

$$\mathrm{v}(x) \approx 1 - x + \left(\frac{1}{6}\right)x\ln x. \tag{A7}$$

**Table 4.** Constants for use in connection with equations (A 5) and (A 6).

| $i$ | $p_i$ | $q_i$ | $s_i$ | $t_i$ |
|---|---|---|---|---|
| 0 | — | — | 0.0532499727 | 0.187 5 [ = 3/16] |
| 1 | 0.03270530446 | 0.1874993441 | 0.02422225959 | 0.03515555874 |
| 2 | 0.009157798739 | 0.01750636947 | 0.01512205958 | 0.01912752680 |
| 3 | 0.002644272807 | 0.005527069444 | 0.007550739834 | 0.01152284009 |
| 4 | 0.000089871738 11 | 0.001023904180 | 0.000639172865 9 | 0.003624569427 |
| 5 | — | — | −0.000048819745 89 | — |

$u(1) = 3\pi/8\sqrt{2} \cong 0.8330405509$

**Table 5.** The maximum magnitudes of the errors in the F06 formulae for $v(x)$ and $t_1(x)$, in relevant ranges of $x$.

| $v(x)$—exact ranges | | $v(x)$—errors in 'F06 formula' | |
|---|---|---|---|
| $x$-range | $v$-range | max. {|error|} | max. {|%error|} |
| $0 \le x \le 1$ | $1.0000 \ge v \ge 0.0000$ | 0.0024 | 0.33% |
| $0.15 \le x \le 0.45$ | $0.8002 \ge v \ge 0.4886$ | 0.0024 | 0.33% |
| $t_1(x)$—exact ranges | | $t_1(x)$—errors in 'F06 formula' | |
| $x$-range | $t_1$-range | max. {|error|} | max. {|%error|} |
| $0 \le x \le 1$ | $1.0000 \le t_1 \le 1.1107$ | 0.0041 | 0.39% |
| $0.15 \le x \le 0.45$ | $1.0070 \le t_1 \le 1.0378$ | 0.0041 | 0.38% |

The equivalent F06 formula for $t_1(x)$, derived from (A 7) via definitions (A 2) and (A 3) above, is

$$t_1(x) \approx 1 + \frac{x}{9} - \left(\frac{1}{18}\right)x \ln x. \tag{A 8}$$

Maximum errors in these F06 formulae are listed in table 5. As shown in [45], when considered over the wide range $0 \le x \le 1$, these F06 formulae are more accurate than all older approximations of equivalent complexity. The success of these F06 formulae presumably arises because the F06 formula for $v(x)$ mimics the form of the lowest terms in the exact series expansion [44].

## A.4. The Elinson–Shrednik approximation

One of the most effective of the older approximations is the Elinson-Shrednik (ES) approximation (see equation (6.10) on p. 168 of [46]). This formula has been widely used in older Russian-language literature, and in many contexts is a useful approximation. The ES approximation has the form

$$v(x) \approx 0.95 - 1.03x \tag{A 9}$$

and

$$t_1(x) \approx \sqrt{(1.1)} \approx 1.05. \tag{A 10}$$

## A.5. The Murphy–Good temperature correction factor $\lambda_T$

An expression for the temperature correction factor $\lambda_T$ in equation (1.1) was first developed by Murphy & Good [4]. In the Swanson & Bell notation [47], $\lambda_T$ is written

$$\lambda_T = \frac{p\pi}{\sin(p\pi)}, \tag{A 11}$$

and the Swanson–Bell parameter $p$ is estimated by the ratio

$$p \approx \frac{k_B T}{\delta_F^{SN}}, \tag{A 12}$$

where $k_B T$ is the Boltzmann factor, and the parameter $\delta_F^{SN}$ (called here the *barrier-strength decay-width, at the Fermi level, for the SN barrier*) is obtained as follows. In the simple-JWKB tunnelling formalism, the

tunnelling probability $D$ for an SN barrier of zero-field height $H$ can be written

$$D \approx \exp[-G^{SN}] = \exp\left[\frac{-v_H b H^{3/2}}{F_L}\right], \quad (A\,13)$$

where $G$ is called here the *barrier strength*, but is also known as the 'Gamow factor'; $G^{SN}$ is the barrier strength for the SN barrier; and $v_H$ is the value of $v(x)$ that applies to an SN barrier defined by $H$ and $F_L$. The parameter $\delta_F{}^{SN}$ is then given by

$$1/\delta_F{}^{SN} \equiv \left(\frac{\partial G^{SN}}{\partial H}\right)\Big|_{H=\phi} = \left(\frac{3}{2}\right)t_F b \phi^{1/2} F_L{}^{-1}, \quad (A\,14)$$

where, as before, $t_F$ is the value of $t_1(x)$ that applies to an SN barrier defined by $\phi$ and $F_L$.

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
