## [Peer Review File · Royal Society Open Science]

Review History

RSOS-201986.R0 (Original submission)

Review form: Reviewer 1

Is the manuscript scientifically sound in its present form?

Yes

Are the interpretations and conclusions justified by the results?

Yes

Is the language acceptable?

Yes

Do you have any ethical concerns with this paper?

No

Have you any concerns about statistical analyses in this paper?

No

Recommendation?

Accept with minor revision (please list in comments)

Comments to the Author(s)

This article reports that a comparison of theoretically and experimentally derived k-parameter, present on an empirical equation for the measured current vs. measured voltage, could be a useful tool for comparing field electron emission theory and experiment. In addition, the authors showed that extracted k-values, from numerical solution of Laplace equation and using a planar transmission approximation for emitted current calculations, are sensitive to electron emission theory used, depending also on geometry of the emitter. This is very interesting theoretical work, with potential implications on providing interpretations of "precise" current-voltage characteristics of ideal field emitter devices. The language is clear and rigorous. This manuscript merits publication in Royal Society Open Science, but the authors are requested to respond comments below.

- This reviewer would like to know more precisely from authors, if possible at the present stage of knowledge, about their expectations of experimental procedures to confirm the main proposal of this work, namely: using k-parameter to validate their claim by using reliable experiments with metal LAFEs. This question is justified since the authors reported, for instance, that the local-gradient method met problems in well established experimental FE current-voltage characteristics from metals. In other words, please, clarify more precisely the meaning of the term "high-quality experimental results" on page 5, and how this eventually would be possible at present.

- The authors searched for a k-value that provided a best linearity according to the criterion related to the sum of least square deviations. This looks a plausible technique for this reviewer. My question is: Does the best linearity, by using the local gradient method, implies that a characterization parameter of an emitter could be reliably extracted, as compared with that carefully calculated from first-principles electrostatic simulations? Please, clarify it.

-The authors used the well-known "hemisphere on a cylindrical post" (HCP) model to classically represent also a multi-walled carbon nanotube. This reviewer would expect a more clear relation between small radius HCP model and single-walled CNTs. Please, could the authors comment about that?

- A small typo: there is an extra ")" at the final of the penultimate paragraph on pg. 2.

Review form: Reviewer 2

Is the manuscript scientifically sound in its present form?

Yes

Are the interpretations and conclusions justified by the results?

Yes

Is the language acceptable?

Yes

Do you have any ethical concerns with this paper?

No

Have you any concerns about statistical analyses in this paper?

No

Recommendation?

Accept with minor revision (please list in comments)

Comments to the Author(s)

See attached file (Appendix A).

Decision letter (RSOS-201986.R0)

Dear Dr Forbes

On behalf of the Editors, we are pleased to inform you that your Manuscript RSOS-201986 "The pre-exponential voltage-exponent as a sensitive test parameter for field emission theories" has been accepted for publication in Royal Society Open Science subject to minor revision in accordance with the referees' reports. Please find the referees' comments along with any feedback from the Editors below my signature.

Please submit your revised manuscript and required files (see below) no later than 7 days from today's (ie 03-Feb-2021) date. Note: the ScholarOne system will 'lock' if submission of the revision is attempted 7 or more days after the deadline. If you do not think you will be able to meet this deadline please contact the editorial office immediately.

on behalf of Dr Robert Young (Associate Editor) and Miles Padgett (Subject Editor)
openscience@royalsociety.org

Associate Editor Comments to Author (Dr Robert Young):

Associate Editor: 1

Comments to the Author:

Please consider the detailed comments made by the two reviewers. They have both suggested some minor revisions be made to the manuscript, we'd be grateful if you could follow their advice before returning a revised manuscript for publication.

Reviewer comments to Author:

Reviewer: 1

Comments to the Author(s)

This article reports that a comparison of theoretically and experimentally derived k-parameter, present on an empirical equation for the measured current vs. measured voltage, could be a useful tool for comparing field electron emission theory and experiment. In addition, the authors showed that extracted k-values, from numerical solution of Laplace equation and using a planar transmission approximation for emitted current calculations, are sensitive to electron emission theory used, depending also on geometry of the emitter. This is very interesting theoretical work, with potential implications on providing interpretations of "precise" current-voltage characteristics of ideal field emitter devices. The language is clear and rigorous. This manuscript merits publication in Royal Society Open Science, but the authors are requested to respond comments below.

- This reviewer would like to know more precisely from authors, if possible at the present stage of knowledge, about their expectations of experimental procedures to confirm the main proposal of this work, namely: using k-parameter to validate their claim by using reliable experiments with metal LAFEs. This question is justified since the authors reported, for instance, that the local-gradient method met problems in well established experimental FE current-voltage characteristics from metals. In other words, please, clarify more precisely the meaning of the term "high-quality experimental results" on page 5, and how this eventually would be possible at present.

- The authors searched for a k-value that provided a best linearity according to the criterion related to the sum of least square deviations. This looks a plausible technique for this reviewer. My question is: Does the best linearity, by using the local gradient method, implies that a characterization parameter of an emitter could be reliably extracted, as compared with that carefully calculated from first-principles electrostatic simulations? Please, clarify it.

-The authors used the well-known "hemisphere on a cylindrical post" (HCP) model to classically represent also a multi-walled carbon nanotube. This reviewer would expect a more clear relation between small radius HCP model and single-walled CNTs. Please, could the authors comment about that?

- A small typo: there is an extra ")" at the final of the penultimate paragraph on pg. 2.

Reviewer: 2

Comments to the Author(s)

See attached file

===PREPARING YOUR MANUSCRIPT===

===PREPARING YOUR REVISION IN SCHOLARONE===

Author's Response to Decision Letter for (RSOS-201986.R0)

See Appendix B.

Decision letter (RSOS-201986.R1)

Dear Dr Forbes,

It is a pleasure to accept your manuscript entitled "The pre-exponential voltage-exponent as a sensitive test parameter for field emission theories" in its current form for publication in Royal Society Open Science.

Best regards,

on behalf of Dr Robert Young (Associate Editor) and Miles Padgett (Subject Editor)
openscience@royalsociety.org

Appendix A

Jour: RSOS-201986

Auth: Forbes, Richard; Popov, Eugeni; Kolosko, Anatoly; Filippov, Sergei

Titl: The pre-exponential voltage-exponent as a sensitive test parameter for field emission theories

Commonly used equations of field emission current I use $I(F) = CF^2 \exp[-b/F]$ (after Eq. (5) of Ref. [1] for zero temperature), a form similar to the formulation of Murphy and Good [2] for current density J . The authors *correctly* note that there are processes that alter the power that F is raised to from 2 to a more general k , as a consequence of experimentally *expected* complications such as geometry, adequacy of the approximation to the transmission probability, and the consequences of non-zero temperature of the emitter. The temperature problem is known to be increasingly acute for fiber, wire, and nanotube emitters. Statistics could be mentioned. The authors argue that a serious theoretical investigation of what factor of k best accounts for expected variations due to the complications they enumerate (as the authors as well as others have been proactive in advocating). That argument is important to this reviewer, and efforts to bring it to the attention of the community is endorsed. Simple theoretical models that show the consequences of the complications in great clarity, and extend those findings to increasingly accurate experimental studies, as the authors do, is good: doing so may enable greater understanding of fundamental physical phenomena that can be teased thereby out of data, as argued by the authors. The methods behind the analysis are careful and justified. It is a strength, not a weakness, to focus on simple models (*e.g.*, hemisphere on post) so as to show the consequences of geometry of a single emitter: doing so unequivocally demonstrates how properties associated with physical emitters might be expected to impact the theory: in fact, the analytical models are a better measure of the accuracy of the methods than comparisons to experimental data (with their related ambiguities) would be. The lead author, in particular, has been proactive in making field emission theory serve the needs of the experimental community: the present work contributes to that agenda. The writing is precise and clear. The justifications for the conclusions drawn are careful and well supported. The simulations showing the impact of geometry on k quite clearly demonstrate the importance of the argument for HCP emitters. References to the literature are appropriate. The content is well suited to the readership of RSOS, as is generally true for theoretical studies of field emission. The manuscript merits publication.

I invite the authors to consider some comments and/or recommendations first.

1. Content in the Introduction is also found in Refs. [3] and [4], which gives greater detail. Some editing for conciseness appears achievable for pages 3 and 4 in particular.
2. The citation to Ref. [39] on page 26, line 26 appears to be mistaken. I believe the authors meant Ref. [42].
3. Lines 35-39, page 6: the value of k is affected by what they call LECD as well as the surface shape of the emitter. Other contributions are not mentioned, such as the statistical variation of emitter properties like work function and emitter radius. At very high fields, the adequacy of the exponential approximation of D_{Fa} (their Eq. 2.2d) compared to the Kemble form (see Ref. [2], which the authors are intimately aware of) will increasingly affect k . Temperature effects are mentioned (λ_T), but minimized by restricting attention to room temperature as an upper limit, whereas nanotube emitters can run hot. These merit mention in passing.
4. Procedural Conclusion (7) on page 21 asserts that “...*there will be no merit, in future sim-*

ulations, in using anything other than the high-precision formula for $v(x)$ given in Appendix A.” This is hyperbolic. It is *desirable* to use high precision formulations on general principle (doing otherwise is sloppy), but as a pragmatic matter, the energy dependent coefficients of the transmission probability, the non-usage of the Kemble approximation, the variations in both work function and local microscopic surface features all conspire to make insistence on high precision counterproductive for computational reasons, *especially given the significant accuracy* of the authors’ own “simple good approximations” that meet need. Simplicity competes with accuracy in this case. I feel simplicity wins decisively, but acknowledge that the authors may differ.

5. The last paragraph on page 21, and its conclusion on page 22, contain matters of great importance. For example, the authors observe “...*the simple-JWKB approximation is valid only if the relevant component of the electron wave-function in effect obeys the one-dimensional Schrödinger equation as found when separation is made in Cartesian coordinates.*” and next discuss coherent emission effects. These complications, to this reviewer, are profoundly important theoretical direction that merits greater prominence in the manuscript: will the authors consider moving that discussion to the Introduction?
6. Appendix A.4 (The Elinson-Shrednik (ES) approximation) is a compact approximation for experimentalists, but lacking in comparison to Appendix A.3. Why is it included? The authors have, in the previous item, asserted the importance of high precision that this appendix subverts. There are recent approximations to the Gamow factor G^{SN} in the literature that are quite accurate and do not use the “field emission special mathematical functions” (but have a relation to them). If alternates are useful, why not them?
7. Appendix A.5 concerns thermal effects, but numerous studies give better approximations than Eq. (A.11) to treat temperature. λ_T does not figure significantly in the manuscript and is a dated approximation. A more recent version would be greater utility if it is needed at all.

References

- [1] R.A. Millikan, and C.C. Lauritsen, “Relations of field currents to thermionic currents”, Proceedings of the National Academy of Sciences of the United States of America **14**(5-6), 445-49 (1928).
- [2] E.L. Murphy, and R.H. Good, “Thermionic Emission, Field Emission, and the Transition Region”, Phys. Rev. **102**(6), 1464-1473 (1956).
- [3] R.G. Forbes, “Comments on the continuing widespread and unnecessary use of a defective emission equation in field emission related literature”, J. Appl. Phys. **126**(21), 210901 (2019).
- [4] R.G. Forbes, and J.H.B. Deane, “Reformulation of the standard theory of Fowler-Nordheim tunnelling and cold field electron emission”, Proc. R. Soc. A **463**(2087), 2907-2927 (2007).

Appendix B

RESPONSE TO REVIEWERS

GENERAL COMMENTS

R1. We are very grateful to both reviewers for their careful comments and questions. In particular, these have caused us to think more critically about the purpose of this paper, and about the need to describe this purpose with sufficient care. As a result, we have slightly modified the introduction, have added significantly to Section 6, and have added a few explanatory comments in other parts of the paper. Responses to individual reviewer comments are made below, and are numbered R1, etc. As requested, a file showing material inserted is provided.

Our basic thinking about this paper is now as follows.

This paper is the **start** of a process to get more science into field electron emission (FE). (By this we mean we want to be able to make better and more decisive quantitative comparisons between theory and experiment.) We expect it may be the first of a series of papers, by ourselves and (probably) other authors.

This first paper is seen as a "basic proof of concept". The concept is that the pre-exponential voltage exponent k in the so-called "empirical FE equation" can in principle be used as a relatively sensitive test of compatibility between theory and experiment, if sufficiently sensitive experimental procedures can be designed and implemented.

To establish this, we use variants of the Murphy-Good finite-temperature FE equation to generate simulated current-voltage data. This equation is the simplest useful FE equation. It has been around for 60 years, so most researchers with a serious interest in FE theory will be aware of it. Using this equation is sufficient to establish this basic proof of concept.

We do not consider it necessary in the present paper to explore in detail whether and how the methodology suggested here can be applied to the various advanced treatments of FE theory that now exist in the literature. It seems plausible to us that it can be applied to at least some of these treatments, but we leave the details to subsequent papers, by ourselves or other authors.

Likewise, we do not consider it necessary in the present paper to explore in detail how it might be necessary to design experimental apparatus and procedures so that current-voltage measurements "of sufficient quality" can be made.

However, in view of the obvious wish of both reviewers for more information about how the proposed methodology would be implemented, we have restructured and expanded Section 6 of the paper, to give an indication of our thinking about these things.

Detailed responses to reviewers' comments now follow.

REVIEWER 1

Comments to the Author(s)

This article reports that a comparison of theoretically and experimentally derived k -parameter, present on an empirical equation for the measured current vs. measured voltage, could be a useful tool for comparing field electron emission theory and experiment. In addition, the authors showed that extracted k -values, from numerical solution of Laplace equation and using a planar transmission approximation for emitted current calculations, are sensitive to electron emission theory used, depending also on geometry of the emitter. This is very interesting theoretical work, with potential implications on providing interpretations of "precise" current-voltage characteristics of ideal field emitter devices. The language is clear and rigorous. This manuscript merits publication in Royal Society Open Science, but the authors are requested to respond comments below.

R2. We are grateful to the reviewer for his comments on the work.

- This reviewer would like to know more precisely from authors, if possible at the present stage of knowledge, about their expectations of experimental procedures to confirm the main proposal of this work, namely: using k -parameter to validate their claim by using reliable experiments with metal LAFEs. This question is justified since the authors reported, for instance, that the local-gradient method met problems in well established experimental FE current-voltage characteristics from metals. In other words, please, clarify more precisely the meaning of the term "high-quality experimental results" on page 5, and how this eventually would be possible at present.

R3. We have expanded our comments in the discussion section to deal with this point. However, as indicated above, we think more careful discussion of the issues likely to be involved needs a separate paper. We are also exploring experimentally whether it is fact possible that useful progress could be made with the existing experimental facility in the Ioffe Institute FE research group. This facility is sensitively computer controlled, but was designed to measure current-voltage data in order to characterise field emitters working in industrial vacuum conditions. Obviously, such conditions are not ideal for scientific experiments, but it has seemed of interest to attempt such experiments, even if the most likely result is fuller understanding of what the experimental difficulties may actually be. Explorations with single-tip metal emitters, as well as with arrays of (notionally) identical silicon tips, are being carried out.

- The authors searched for a k -value that provided a best linearity according to the criterion related to the sum of least square deviations. This looks a plausible technique for this reviewer. My question is: Does the best linearity, by using the local gradient method, implies that a characterization parameter of an emitter could be reliably extracted, as compared with that carefully calculated from first-principles electrostatic simulations? Please, clarify it.

R4. We are not quite sure that we understand the reviewer's question, so here are two alternative answers.

(a) The value of k that is extracted from a "reliable experiment" will contain entangled information about (i) the emission theory and (ii) the shape of the emitter and/or the dependence of notional emission area on voltage. If, as seems highly likely, we do not initially have reliable information about either of these things, then a "chicken-and-egg" situation exists. If we knew reliably about the emission theory then we could extract information about the effects of emitter shape (etc). Or, if we knew reliably about the effects of emitter shape, then we could possibly deduce information about

emission theory. It seems plausible that we could deduce better information about the effects of emitter shape by carrying out simulations on a wide variety of shapes, and by then collating this information with the several theoretical discussions of effective emission area that do currently exist. Activities toward this end are in progress, by the authors and others. The desirability of further simulation activity of this kind is already explicitly indicated in the paper.

(b) There is a question about the relationship between modern density-functional-theory calculations of (say) carbon nanotubes (CNTs), and finite-element-method analyses of classical conductor shapes (such as the hemisphere-on-cylindrical-post model) that have been used to represent CNTs. Although some progress has recently been made, current emphasis is mainly on comparing the electrostatics. The use of DFT calculations to predict FE current-voltage characteristics is in its infancy. Our view is that it is too early to make reliable statements about whether/how well the methodology suggested in this paper could be used to check predictions about FE current-voltage characteristics derived from DFT theory (or from other sophisticated quantum-mechanical theory), although we certainly hope/expect that this will prove possible in due course. (And we do see encouraging signs, for example work by Lepetit [J. Appl. Phys. 125, 025107 (2019)], that this will in fact prove to be the case.) Probably the most promising situation to investigate (both theoretically and experimentally) would be a defined area of an atomically flat metal surface of known crystallographic orientation.

We consider it to be too early to discuss detailed issues of this kind in the present paper.

-The authors used the well-known "hemisphere on a cylindrical post" (HCP) model to classically represent also a multi-walled carbon nanotube. This reviewer would expect a more clear relation between small radius HCP model and single-walled CNTs. Please, could the authors comment about that?

R6. The historical reason for using the chosen HCP-model shapes in our simulations is that in the work that originally stimulated this paper we hoped to be able to use the theoretical analyses to help us characterise the emitters used in the Ioffe Laboratory experimental work. In the event, our requirement to work in industrial vacuum conditions meant that the experimental results were (certainly at this stage) too noisy to reliably generate consistent k -values. Thus, in the simulations in the present paper, the HCP-model shapes are being used mainly to demonstrate the basic point that the assumption of a non-planar emitter does or can generate a significant contribution to the k -value. Questions as to whether HCP models are good models for real CNTs are, at this point in time, of limited relevance to the present research. (But, in fact, are of great interest to research being coordinated by Prof. T. de Assis at the Federal University of Bahia in Brazil [for example, see C.P. Castro et al., J. Chem. Inf. Model. 60, 714-721 (2020)].) We do expect issues of this general kind to again become relevant to the Ioffe Laboratory work in due course, but not in the immediate future. Again, we consider it to be far too early to discuss issues of this kind in the present paper. However, we would hope that eventually (but not in the near future) it might be possible to use k -value methodology to test DFT theories of CNT field emission against relevant experiments. A significant underlying problem is to find out how to make reliable predictions of transmission probability (and hence of current-voltage characteristics) from DFT models of CNTs. It is not clear to us that, at present, it is reliably understood how to do this.

-
A small typo: there is an extra ")" at the final of the penultimate paragraph on pg. 2.

R7. Thanks, fixed.

REVIEWER 2

1. Content in the Introduction is also found in Refs. [3] and [4], which gives greater detail. Some editing for conciseness appears achievable for pages 3 and 4 in particular.

R8. We have thought carefully about this suggestion. We accept that the introductory material in this article can also be found elsewhere, but nevertheless we continue to consider that it is important to present this material in this paper too. Partly, this is because we want to set out a detailed background here to the simulations carried out later in the paper. Partly, this is because this article is aimed at a wide range of specialists in field electron emission (FE), including those working in experimental FE research. We think that for the latter group it is much better to present a short self-contained account of relevant theory here, rather than expect them to refer to separate largely-theoretical papers. Further, Royal Society Open Science is an "electronic version" only journal. This means that, although conciseness remains a virtue, there isn't quite the same length pressure as with a printed journal. We have added a brief comment to indicate the relevance of equation (1.4).

2. The citation to Ref. [39] on page 26, line 26 appears to be mistaken. I believe the authors meant Ref. [42].

R9. Agreed. Thanks, fixed.

3. Lines 35-39, page 6: the value of k is affected by what they call LECD as well as the surface shape of the emitter. Other contributions are not mentioned, such as the statistical variation of emitter properties like work function and emitter radius.

R9. We agree that we have not been clear enough about procedural matters, and have added a sentence to clarify what thinking lies behind the analysis given. The more general point is made in Section 6 that, in order to reach valid scientific conclusions in a messy theoretical environment, it is necessary to start off by dealing with the simplest relevant situations.

At very high fields, the adequacy of the exponential approximation of DF_a (their Eq. 2.2d) compared to the Kemble form (see Ref. [2], which the authors are intimately aware of) will increasingly affect k .

R10. The reviewer is correct in principle that probably the theory ought to be formulated in terms of the Kemble formalism. We have now introduced an additional equation [new 2.2.d] in order to display this. In practice, we have confined our simulations to the scaled-field (f) range $0.15 \leq f \leq 0.45$ (which is the PASS range of the orthodoxy test), and to temperatures of either (formally) 0 K or 300 K. Following Murphy and Good, and Modinos, usual thinking is that, at these temperatures (for work function $\phi \approx 4.5$ eV), the "simple-JWKB" formalism is adequately valid up to about $f \sim 0.8$. Issues relating to the need to use the Kemble formalism (or a more sophisticated formalism) would not arise when simulations and experiments are performed at or near room temperature, and within the PASS range of the orthodoxy test. We have added a sentence to note this.

Temperature effects are mentioned (LT), but minimized by restricting attention to room temperature as an upper limit, whereas nanotube emitters can run hot. These merit mention in passing.

R11. For a "basic demonstration of concept" it is sufficient and best to confine discussion to temperatures near room temperature, or below, and this is now indicated in the paper. Later, one might investigate emitters that are "externally heated". We are certainly aware that nanotubes can "run hot". However, if this is the result of resistive heating and/or Nottingham-style heating, then it is possible that the measured current-voltage characteristics would not accurately reflect the emission characteristics but would also involve resistivity (etc.) effects that could give rise to voltage drop along the emitter (and perhaps to current dependence in field enhancement factors). In this case the experimental results might be of limited value for scientific experiments intended to test emission theory.

In fact, we do now have simulation data for a temperature of 900 K, and this appears to show that k has risen significantly (more than doubled). It is possible that the proposed least-residual and local-gradient techniques would turn out to be quite sensitive to temperature effects. However, to keep things simple, we do not report this in the paper.

4. Procedural Conclusion (7) on page 21 asserts that "...there will be no merit, in future simulations, in using anything other than the high-precision formula for $v(x)$ given in Appendix A." This is hyperbolic. It is desirable to use high precision formulations on general principle (doing otherwise is sloppy), but as a pragmatic matter, the energy dependent coefficients of the transmission probability, the non-usage of the Kemble approximation, the variations in both work function and local microscopic surface features all conspire to make insistence on high precision counterproductive for computational reasons, especially given the significant accuracy of the authors' own "simple good approximations" that meet need. Simplicity competes with accuracy in this case. I feel simplicity wins decisively, but acknowledge that the authors may differ.

R12. We take the point that in some circumstances considerations of numerical efficiency might favour use of the "simple good approximation" rather than the "high precision approximation", and have now inserted this point into the paper. In reality—possibly because of the relative simplicity of the models analyzed so far—we have not encountered circumstances of this type. In practice, the related finite-element-method calculations seem to have been more demanding of computer resources.

On the more general issues raised by the reviewer, we think the overall procedural situation is more complicated than the reviewer has realized. The question at the heart of this paper is whether some parameter or feature inserted into the simulations is reflected in and can be extracted from the "best k -value" extracted from the data-analysis methods. One can arrange the simulations in order to avoid some or most of the complications mentioned by the reviewer, and in these circumstances we stand by our view that there is limited merit in using anything other than the high-precision approximation.

There then needs to be a separate complicated discussion (in one or many separate papers) of how to set up the most appropriate scientific experiments, or alternatively of the merits of different forms of experiment. Hopefully, one can set up experiments (possibly very different from the usual arrangements) that avoid some or most of the complications mentioned by the reviewer, which we are well aware of.

It may be that relatively simple experiments using existing apparatus, and carefully carried out, can contribute to some issues where predicted differences in k are large, in particular to the desirability of providing a convincing experimental proof that the Murphy-Good FE equation is better physics than the original 1928 Fowler-Nordheim FE equation. We have already taken steps towards exploring this issue [see: Popov E.O., Kolosko A.G., Filippov S.V., A Test for the Applicability of the Field Emission Law to Studying Multitip Field Emitters by Analysis of the Power Index of the Preeponential Voltage Factor, Tech. Phys. Lett., v.45, 2019, p. 916-919. DOI: <http://dx.doi.org/10.1134/S106378501909027X>].

5. The last paragraph on page 21, and its conclusion on page 22, contain matters of great importance. For example, the authors observe "...the simple-JWKB approximation is valid only if the relevant component of the electron wave-function in effect obeys the one-dimensional Schrodinger equation as found when separation is made in Cartesian coordinates." and next discuss coherent emission effects. These complications, to this reviewer, are profoundly important theoretical direction that merits greater prominence in the manuscript: will the authors consider moving that discussion to the Introduction?

R13. We are somewhat reluctant to give these two particular issues prominence at the start of the paper, for several reasons. Partly, this is because one of us (RGF) considers that they are only two of several issues of fundamental importance that affect FE theory (and possibly affect tunnelling theory more generally). Partly, this is because we think that it is better to mention them in the discussion section, after the basic demonstration that the pre-exponential voltage exponent is sensitive to FE theoretical details. Partly, this is because at this point in time we have no good knowledge of how the value of k might be affected. Partly this is because it seems probable that related changes in k might be relatively small; thus testing this kind of theoretical issue probably needs to wait until after larger effects have been investigated.

However, we certainly are in favour of having these issues discussed in principle in a separate paper.

6. Appendix A.4 (The Elinson-Shrednik (ES) approximation) is a compact approximation for experimentalists, but lacking in comparison to Appendix A.3. Why is it included? The authors have, in the previous item, asserted the importance of high precision that this appendix subverts. There are recent approximations to the Gamow factor GSN in the literature that are quite accurate and do not use the field emission special mathematical functions" (but have a relation to them). If alternates are useful, why not them?

R14. There are three separate issues here: (1) Why was the Elinson-Shrednik ("ES") approximation included in the simulations? (2) Why is it recorded in an Appendix? (3) Why did we not investigate modern alternative formulations of FE theory, for example those that are not directly based on the Nordheim 1928 idea of a correction factor to the results for an exactly triangular barrier?

The short answer to the first two issues is "history", namely the slightly different histories of FE theory in Western countries and in the former Soviet Union (and now in the Russian Federation).

The "ES" approximation for " v " (normally denoted by θ in Russian-language literature) was first clearly published in a Russian-language textbook in 1974 and has been extensively used in Russian-language literature from then until now. Since this paper is the outcome of a Russian-UK collaboration, it was only natural that this approximation should be investigated. The ES approximation is one of the best of the two-term approximations for " v " but is very poorly known in English-language literature. In fact, we are not aware of any English-language paper that states the ES approximation explicitly and clearly. This is the reason why it is recorded explicitly in Appendix A4. However, it has been part of our conclusions that there is limited merit in using this approximation in further exploratory simulations, and we have now modified the conclusions to make this point explicitly.

By contrast, comparison of the "simple good approximation" with 11 earlier approximations (not including the ES approximation) has already been carried out [Forbes & Deane, JVSTB 28, C2A33 (2010)]. Although some of these yield better results over limited ranges of f , the simple good approximation (which is labelled "F06" in the cited paper) is the "best general-purpose simple approximation". Hence there seems no need to consider, in this "proof of concept" paper, any of these other approximations.

On the issue concerning modern alternative formulations, the same issue about this being a "BASIC proof of concept paper" is relevant. We think that experimentalists are much more likely to understand a paper that uses different variants of the Murphy-Good FE equation, and different approximations for " v ", than they are to understand a paper that uses advanced modern treatments of tunnelling theory to make basically the same point. Hence advanced modern treatments are not discussed in detail in the paper, though we do mention two of a larger number of issues.

7. Appendix A.5 concerns thermal effects, but numerous studies give better approximations than Eq. (A.11) to treat temperature. \$\lambda_T\$ does not figure significantly in the manuscript and is a dated approximation. A more recent version would be greater utility if it is needed at all.

R16. Our expression for the factor λ_T is the standard Swanson and Bell form for the mathematical treatment used by Murphy and Good to develop their finite-temperature field emission equation. One of us (in unpublished work with J.H.B. Deane) has verified by an independent mathematical method that the mathematical result is correct, provided that certain mathematical assumptions are adequately valid (which looks reasonably so at room temperature).

This factor λ_T was included in the room-temperature calculations as a matter of completeness, but surprised us by the size of the effect that the standard expression for λ_T had on the value of k , even at room temperature. The implications are: (1) the temperature correction factor does need to be included in the theoretical discussion; and (2) experiments to test "field effects" in FE theory should **not** be done at elevated temperature and (in some contexts) might be better done at a well-defined lower temperature (e.g., liquid nitrogen temperature). This point is now mentioned in Section 6(b).

At this stage of methodology development, both theory and experiment are being confined to room temperature and below, and we plan to operate within the validity limits of the derivation of the Murphy-Good finite temperature FE equation. Our view is that in these circumstances the standard Swanson and Bell expression for λ_T is fit for purpose. Our understanding is that the more sophisticated modern treatments of temperature effects in thermal-field electron emission, to which the reviewer refers, do reduce or should reduce to the Swanson and Bell form at room temperature and below, provided that we operate within the validity limits of the derivation of the Murphy-Good finite temperature FE equation.

RG Forbes,
on behalf of the author team
15 February 2021